# Temperature relaxation in strongly-coupled binary ionic mixtures

R. Tucker Sprenkle 1,3, L. G. Silvestri2, M. S. Murillo 2✉ & S. D. Bergeson 1✉

New facilities such as the National Ignition Facility and the Linac Coherent Light Source have pushed the frontiers of high energy-density matter. These facilities offer unprecedented opportunities for exploring extreme states of matter, ranging from cryogenic solid-state systems to hot, dense plasmas, with applications to inertial-confinement fusion and astrophysics. However, significant gaps in our understanding of material properties in these rapidly evolving systems still persist. In particular, non-equilibrium transport properties of strongly-coupled Coulomb systems remain an open question. Here, we study ion-ion temperature relaxation in a binary mixture, exploiting a recently-developed dual-species ultracold neutral plasma. We compare measured relaxation rates with atomistic simulations and a range of popular theories. Our work validates the assumptions and capabilities of the simulations and invalidates theoretical models in this regime. This work illustrates an approach for precision determinations of detailed material properties in Coulomb mixtures across a wide range of conditions.

[1] Department of Physics and Astronomy, Brigham Young University, Provo, UT 84602, USA. [2] Department of Computational Mathematics, Science and Engineering, Michigan State University, East Lansing, MI 48824, USA. [3] Present address: Honeywell Quantum Solutions, 303 S Technology Ct, Broomfield, CO 80021, USA. ✉email: murillom@msu.edu; scott.bergeson@byu.edu

Advancing the frontier of dense plasma science requires a deep understanding of plasma processes in extreme and transient conditions. Accurately modeling high energy-density plasmas (HEDP) requires detailed and reliable models of collective phenomena such as continuum depression heating[1], turbulence and mixing[2–4], diffusion[5,6], viscosity[7], and many other physical processes[8,9]. These transport processes are critical components of modeling codes for laser-driven plasmas[10] and stellar atmospheres[11,12].

Predicting transport coefficients in plasmas dominated by strong dynamical collision processes remains an unresolved issue[13]. Plasmas are considered strongly coupled when the ion coupling parameter Γ, defined as the ratio of the nearest-neighbor electrical potential energy to the average kinetic energy, is larger than 1. For plasma mixtures, the coupling parameter of species $\alpha$ is defined as

$$\Gamma_\alpha = \frac{(Z_\alpha e)^2}{4\pi\epsilon_0 a_{\mathrm{ws}}} \frac{1}{k_{\mathrm{B}} T_\alpha}, \tag{1}$$

where $Z_\alpha$ is the ion charge number, $T_\alpha$ is the temperature, and $a_{\mathrm{ws}} = (3/(4\pi n_{\mathrm{tot}}))^{1/3}$ is the average distance between ions, $n_{\mathrm{tot}} = \sum_\alpha n_\alpha$ is the total ion density. Strongly coupled plasmas are characterized by large-angle scattering with tight particle correlations and dynamical screening. The characteristic time scales for collisions and collective mode periods overlap, clouding the otherwise clear separation that typically simplifies theoretical models. This overlap is expected to be important to ion–ion thermal decoupling where the ion mass ratio is close to unity, as measured in a recent experiment[14].

The challenge in plasma theory is that when $\Gamma \gtrsim 1$, standard kinetic and hydrodynamic approximations are not entirely appropriate. For example, cross-sections in the Boltzmann equation accurately describe transport in plasmas characterized by binary ionic collisions[15]. In strongly coupled plasmas, binary collisions are important, but they do not exclusively describe all the ion-ion interactions. Conversely, dielectric functions in the Lenard–Balescu equation appropriately describe transport when collisions are characterized by weak many-body scattering events[16,17]. When neither of these two limits is realized, hybrid models are required[18]. One approach is to build the many-body screening into an effective potential and to use it when computing cross-sections, thereby capturing the strengths of both limits[15,19,20]. Carefully designed and accurately diagnosed laboratory experiments are required to test the reliability of these hybrid approaches[21]. One such experiment is presented in this paper.

Because most plasmas are created out of equilibrium, understanding temperature relaxation is critical for modeling the evolution of multi-temperature HEDPs[22–27]. Temperature relaxation has been studied extensively for electron-ion systems[16,17,28–35]. However, most plasma theories are tailored for the case of widely disparate mass (electrons and a single ion species). These theories have been compared to MD simulations with varying degrees of success[30,35]. However, explicit electron-ion MD simulations often rely on quantum statistical potentials[36,37] which may only be valid in thermodynamic equilibrium[29]. This complicates comparisons of MD simulations with theory because disagreements can be attributed to uncertainties in the interaction potentials instead of theoretical models.

At a fundamental level, it is appropriate to ask when and if a two-temperature system can form. Simply mixing hot and cold particles together may result in a non-Maxwellian velocity distribution function in which no true "temperature" is defined. In mixtures of particles with different masses, it is possible for each mass species to have an independently defined Maxwellian velocity distribution and, therefore, temperature. When the species' temperatures evolve slowly enough that the Maxwellian distributions are maintained, temperature relaxation becomes a meaningful concept. Electron-ion plasmas[22,31,38,39] and electron–hole plasmas[40–42] are binary systems in which two temperatures are well established. However, understanding the basic physics of temperature relaxation, therefore, requires the ability to vary the mass ratio and to cleanly measure the time-evolving temperatures.

Ultracold neutral plasmas (UNPs) provide an idealized platform for measuring plasma transport properties[9,43–46]. Recent laboratory experiments have shown UNPs to be effective HEDP simulators over a limited range of parameters[9,21,47–53]. Both UNPs and HEDPs can be described using dimensionless parameters. One of them is Γ, which involves ratios of temperature, charge, and density. Another is the Knudsen number, which involves a ratio of the ion mean free path divided by the characteristic length scale. The values of these dimensionless parameters are similar in both HEDPs and UNPs[54]. Therefore, UNP experiments can probe some interaction physics relevant to HEDP systems.

UNPs are strongly-coupled, non-degenerate, quasi-homogeneous, quasi-steady-state plasmas in which the charge state is well-known. The initial electron temperature is independent of the ion temperature, and it is chosen with sub-percent accuracy. The time-evolving temperatures and densities of each ion species are readily and simultaneously determined using identical techniques for each species. Furthermore, the equation of state for the electrons is well-known, dramatically reducing the complexity of interpreting experimental data and applying plasma models[55]. As HEDP simulators, UNPs test collision physics without the complications of high density, inaccessibly short time scales, high transient pressures, quantum potentials, and extreme optical opacity.

In this paper, we report measurements of the velocity field evolution in an expanding dual-species UNP. This allows us to characterize both the hydrodynamic flow as well as the velocity distribution functions for each species. We report hydrodynamic flow-velocity locking as the plasma expands. We perform an analysis of the velocity distributions, confirming the establishment of a two-temperature quasi-equilibrium system. The experiment, coupled with computer simulations, allows us to meaningfully measure the ion-ion temperature relaxation rate in a two-temperature dual-species plasma[13,21]. These results further validate that uniform density MD simulations provide high fidelity distribution functions for determining temperature evolution and transport properties. Our recent work points to the importance of coupled modes in ion-ion relaxation processes[13]. However, we did not demonstrate the establishment of a multi-temperature system, the existence of local thermodynamic equilibrium, or justify the use of uniform-density molecular dynamics (MD) simulations. These critical questions are addressed in the present work.

The dual-species UNP system allows precise control of the ion mass ratio and plasma stoichiometry. We show that within the experimental uncertainties, the measured temperature relaxation rates match the results of classical MD simulations. The same sign of charge removes ambiguities in the choice of the ion–ion interaction potential. We compare rates extracted from these simulations with theoretical predictions in the few cases where the mass ratio dependence can be readily identified. We find reasonably good agreement with a recent model based on an effective Boltzmann equation[15]. This work establishes the use of dual-species UNPs as a unique platform for studying ion transport properties in the strongly coupled plasma regime.

## Results

**Dual-species ultracold neutral plasmas**. Our dual-species UNP is formed by photo-ionizing laser-cooled Ca and Yb atoms in a magneto-optical trap (MOT)[9,21]. The spatial density profile of the trapped neutral atoms is approximately spherically symmetric and Gaussian. To a good approximation, the spatial density profile is described by the function $n = n_0 \exp[-r^2/(2\sigma_0^2)]$. In our experiments, the initial rms sizes of the Ca and Yb atomic clouds $\sigma_0$ ranges from 300 to 1000 μm. The peak density $n_0$ of Ca and Yb ranges from 0.1 to $3.0 \times 10^{10}$ cm$^{-3}$, depending on the MOT parameters. The temperature of the neutral atoms in the trap is around 0.002 K.

The ionization process uses ns-duration laser pulses to ionize Ca and Yb atoms at threshold, as described in the Methods section. This produces a very cold, metastable, out-of-equilibrium plasma with separate temperatures for the electrons and each ion species. The electron temperature is determined by the photo-ionization laser wavelengths. However, the ion temperatures are determined by their mutual interaction after ionization. Typical values are $T_e$ ranging from 10 to 1000 K and $T_i$ in the range of 1–2 K, depending on the initial plasma density.

The density evolution in our dual-species UNP is shown in Fig. 1. For this measurement, the initial rms size of the neutral Ca atomic cloud ($\sigma_0 = 0.76$ mm) was chosen to be larger than the Yb atomic cloud ($\sigma_0 = 0.44$ mm). The peak plasma densities are $n_0^{Ca} = 1.4 \times 10^{10}$ cm$^{-3}$ and $n_0^{Yb} = 2.7 \times 10^{10}$ cm$^{-3}$. The ions are singly-ionized ($Z = 1$) and the electron temperature is 96 K. As the plasma evolves, the larger mass and higher density of the Yb$^+$ ions ($m_{Yb} = 174$ a.m.u.) prevents the central portion of the Ca$^+$ ion density distribution ($m_{Ca} = 40$ a.m.u) from expanding. This "frictional" confinement of the Ca$^+$ ions occurs because the two species are strongly coupled together where the Yb$^+$ density is high.

As the plasma expands, the consequences of this confinement become apparent. Where the Yb$^+$ density gradient is high, the lighter Ca$^+$ ions are accelerated radially outward. In the cold plasma approximation, the momentum equation for a mixture of ions can be written as

$$\frac{\partial \mathbf{u}_\alpha}{\partial t} = -\frac{k_B T_e}{m_\alpha} \frac{\nabla n_e}{n_e} - \sum_{\beta \neq \alpha} \nu_{\alpha\beta}^m (\mathbf{u}_\alpha - \mathbf{u}_\beta), \quad (2)$$

where $\nu_{\alpha\beta}^m$ is the momentum relaxation collision frequency and $\mathbf{u}_\alpha$ is the hydrodynamic velocity of ion species $\alpha = \{Ca, Yb\}$. Details for these equations are given in the Methods section. Some of the Ca$^+$ ions find themselves between distant, hotter electrons and the heavier Yb$^+$ ions. The $\nu_{\alpha\beta}^m$ collision frequency is lower in this region because the Yb$^+$ density is lower, decreasing the friction on the Ca$^+$ relative to the center of the plasma. These Ca$^+$ ions are accelerated outwards more quickly than the ones that are frictionally confined in the center of the Yb$^+$ distribution. Over time, the Ca$^+$ distribution becomes spatially bi-modal as shown in the top row of Fig. 1.

The bottom row of Fig. 1 plots $u_z$, the $z$-component of $\mathbf{u}_\alpha$, for both Ca$^+$ and Yb$^+$ near the center of the plasma at $y = 0$. From very early times in the plasma evolution, the hydrodynamic velocity fields match. Frictional confinement of Ca$^+$ by the heavy Yb$^+$ ions flattens the velocity gradient in the center of the plasma for the first few microseconds. In the center of the plasma, collisions cause $\mathbf{u}_\alpha - \mathbf{u}_\beta \approx 0$.

This observed flow-locking suggests that it is appropriate to model the central region of the plasma as a homogeneous system with zero expansion velocity[56]. Note that this is no longer true at 5 μs. The density variations in our plasmas are not greater than ±10% when considering spatial regions $r < \sigma_0/2$ and for times $t < 0.3(\sigma_0/v_{exp})$ where $v_{exp}$ is a characteristic expansion velocity[56]. The plasmas considered in this paper, $v_{exp}$ is dominated by the heavy Yb$^+$ ions and is typically $v_{exp} = (k_B T_e/m_{Yb})^{1/2} = 67$ m/s. For initial rms Yb$^+$ plasma sizes of $\sigma_0 = 0.72$ and 0.38 mm, for example, evolution times up to 3.2 and 1.7 μs, respectively, are appropriate for the homogeneous plasma model.

In the present work, we will focus on the temperature evolution in the center of the dual-species UNP. Because of the Ca$^+$ and Yb$^+$ mass difference, the initial temperatures of the two ion species are not equal. The heating process which occurs immediately after the plasma is generated asymmetrically deposits kinetic energy into the low-mass Ca$^+$ ions.

**Two-temperature plasmas**. The two-temperature nature of the plasma arises naturally because of the mass difference between the ion species. Although the ions initially retain the mK temperatures of the neutral atom cloud, the ion velocity distributions rapidly broaden as individual ions respond to the sudden appearance of neighboring ions[57]. One way to understand the ion response is to consider the time-evolving pair distribution function, $g(r, t)$. The pair distribution function indicates the probability density of finding a neighboring ion at some distance $r$ at the time $t$. In the neutral atom cloud with essentially no interparticle interactions, $g(r, t)$ is constant everywhere and equal to 1. After ionization, as ions push neighboring ions away, the pair distribution function goes to zero near $r = 0$. Excess electrical potential energy is converted to kinetic energy in this process, dramatically increasing the ion temperature[58,59]. This process is called disorder-induced heating (DIH).

In a dual-species DIH process, the lower mass ions reach a higher temperature. The average kinetic energy of each species of ions in the plasma can be written as

$$k_B T_\alpha(t) = \frac{m_\alpha}{3N_\alpha} \sum_{i=1}^{N_\alpha} v_i^2(t), \quad (3)$$

where the index $i$ indicates the particle of ion species $\alpha$. We propagate the velocity in time using an Euler step, $v_i^2(t) = [\mathbf{v}_i(0) + \mathbf{a}_i(0)t]^2 = v_i^2(0) + 2\mathbf{v}_i(0) \cdot \mathbf{a}_i(0)t + a_i^2(0)t^2$. In a uniform plasma with no spatial order and no bulk flow, the dot

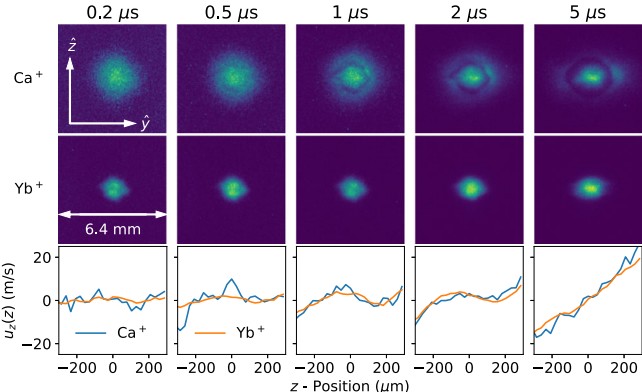

**Fig. 1 Expansion dynamics for a dual-species Ca$^+$/Yb$^+$ UNP.** The top two rows show a vertical cut through the spatial density profile in the center of the plasma. The top row shows the Ca$^+$ density profile and the middle row shows the Yb$^+$ density profile. The bottom row shows $u_z(z)$, the $z$-component of the hydrodynamic velocity $\mathbf{u}_\alpha$ at the time labeled in the figure. The frictional confinement of the Ca$^+$ ions and also the demonstrated hydrodynamic flow locking justify a uniform-density MD simulation as described in the text.

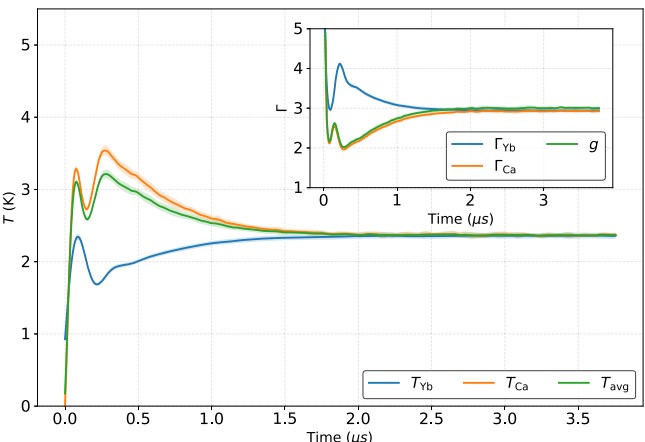

**Fig. 2 Temperature evolution of Ca$^+$/Yb$^+$ mixture.** Data are from MD simulations with Ca$^+$ and Yb$^+$ densities of $4.3 \times 10^9$ and $1.3 \times 10^{10}$ cm$^{-3}$, respectively. For this simulation $\kappa = 0.395$, $T_e = 100$ K, $n_e = 1.73 \times 10^{10}$ cm$^{-3}$. $T_{avg}$ is calculated from Eq. (12), $\Gamma_{Yb,Ca}$ from Eq. (1), and $g$ from Eq. (21). The oscillations in the temperature during the first 0.5 μs are kinetic energy oscillations initiated during the DIH process [see refs. 56 and 58].

product $\mathbf{v}_i(0) \cdot \mathbf{a}_i(0)$ averages to zero, and Eq. (3) simplifies to

$$
\begin{aligned}
k_B T_\alpha(t) &= \frac{m_\alpha}{3N_\alpha} \sum_{i=1}^{N_\alpha} \left[ v_i^2(0) + a_i^2(0)t^2 \right] \\
&= k_B T_\alpha(0) + \frac{t^2}{3N_\alpha m_\alpha} \sum_{i=1}^{N_\alpha} F_i^2(0),
\end{aligned}
\tag{4}
$$

where we have used Newton's second law for the force magnitude $F_i$. This force is due to the electrostatic interaction and is mass independent. As Eq. (4) shows, the smaller mass will reach a higher temperature in the DIH process when the DIH timescale is faster than the thermal relaxation rate.

**MD simulations**. To gain greater insight into plasma dynamics, we perform MD simulations. These are carried out using the Sarkas package, a pure python open-source MD code for non-ideal plasma simulations[60]. UNPs are modeled as a collection of ions interacting via the screened Coulomb (Yukawa) potential,

$$
U(r_{ij}) = \frac{Z_i Z_j e^2}{4\pi\epsilon_0} \frac{1}{r_{ij}} e^{-r_{ij}/\lambda_{TF}}.
\tag{5}
$$

where $i, j$ label the ions in the simulation and $r_{ij}$ their distance. The electrons are not explicitly simulated, but their effect is incorporated in the screening length $\lambda_{TF}$ which is calculated from the electron density and temperature.

Typical MD simulation results are shown in Fig. 2. The temperature of each species is calculated using Eq. (3). The plot clearly shows DIH as the temperature rises during the first few hundred ns. The Ca$^+$ temperature is higher than the Yb$^+$ temperature as predicted in Eq. (4). After the first few μs, the two temperatures approach equilibrium.

Because UNPs are created out of equilibrium[57], we need to establish that a two-temperature system is created according to statistical mechanics. We do this by comparing the ion velocity distributions to a Maxwellian distribution for each species. In our previous work[13], it was suggested that a Hermite polynomial analysis could be used. We show in the Supplementary Information that such an analysis is hampered by its sensitivity to noise in the MD distribution and by a hierarchical relationship between the polynomial coefficients.

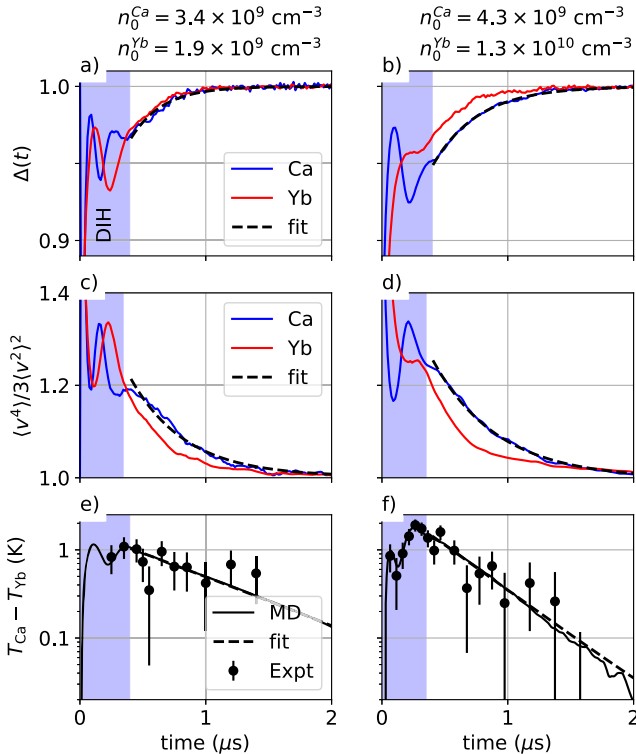

**Fig. 3 Velocity and temperature difference analysis.** Data from two dual-species ultracold neutral plasmas are shown. The top row **a** and **b** plots the fraction of the MD distribution $f(v, t)$ that matches a Maxwellian distribution. In **a**, the Ca and Yb decay time is 250 ns. In **b**, the Ca and Yb decay times are 360 and 270 ns, respectively. The middle row **c** and **d** shows the time evolution of the moment ratio $\langle v^4 \rangle / 3 \langle v^2 \rangle^2$. In **c**, the Ca and Yb moment ratios decay exponentially, with time constants of 440 and 350 ns, respectively. In **d**, the decay constants are 515 and 380 ns. The bottom row **e** and **f** plots the temperature difference between the Ca and Yb ions. The agreement between the experiment (circles) and MD simulations (solid line) is readily apparent. The short vertical black lines represent estimated $1\sigma$ uncertainties in the temperature difference. The exponential temperature relaxation time constants in **e** and **f** are 950 and 450 ns, respectively. The shaded regions indicate the disorder-induced heating (DIH) timescale. After the DIH process is complete, the two-temperature system is well established. The moment ratios are small and tend exponentially toward unity. For all of this data, the electron temperature is 96 K. A correction has been added to **a** and **b** to account for noise in the MD distribution at equilibrium. Data in **e** and **f** are reproduced from Silvestri, L. G. et al., Phys. Plasmas 28, 062302 (2021), with the permission of AIP Publishing.

A straightforward analysis fits the normalized one-dimensional MD distribution $f(v, t)$ to a Maxwellian distribution

$$
\mathcal{M}\left(\frac{v}{\sigma}\right) = \frac{1}{\sigma\sqrt{2\pi}} \exp\left(-\frac{v^2}{2\sigma^2}\right),
\tag{6}
$$

with $\sigma$ as the fit parameter. The integral of the magnitude of the fit residuals are used to determine the fraction of the distribution described by the Maxwellian

$$
\Delta(t) = 1 - \int \left| f(v, t) - M(v/\sigma) \right| dv,
\tag{7}
$$

with $\Delta = 1$ being the perfectly Maxwellian case. A plot of $\Delta(t)$ for two dual-species UNPs is shown in Fig. 3a, b. After the 400 ns duration DIH phase, the distributions are more than 95% Maxwellian.

Deeper insight into the non-Maxwellian nature of the distribution comes from velocity moment ratios. The $n$th velocity moment for a normalized one-dimensional velocity distribution $f(v, t)$ is defined as

$$\langle v^n(t) \rangle = \int dv \; v^n f(v, t). \tag{8}$$

Higher moments are sensitive to anomalies in successively higher velocity ranges.

The moment ratios $<v^4>/3<v^2>^2$ are shown in Fig. 3c, d for a dual-species UNP at two different initial densities. Following the initial DIH phase, after which the two-temperature system is well established, the values of the moment ratios are small and tend monotonically towards the equilibrium value of unity.

The processes of each species equilibrating to a species-dependent temperature while simultaneously relaxing to a single global temperature occur over time scales that depend on species mass and plasma stoichiometry. It is impossible for a two-temperature system to achieve global thermalization unless the distribution is partially non-Maxwellian. For the low-density plasma in Fig. 3a, c, e, species-specific Maxwellianization and moment ratio relaxation occur over time scales of 250 and 400 ns, respectively. The characteristic time for temperature equilibration is 950 ns, as shown in Fig. 3e. This stands in contrast to the higher density plasma in which the processes of Maxwellianization, moment ratio relaxation, and temperature equilibration occur over essentially the same time scale, roughly 400 ns.

**Theoretical considerations**. For the purpose of generalizing these experimental and simulation findings, we consider temperature thermalization using three different plasma theories. For high temperature, low-density plasmas, these theories all agree. However, as the plasmas become colder and/or denser, theoretical predictions diverge. The divergence is entirely due to the treatment of collisions when the plasmas become strongly coupled. A more detailed discussion of the theoretical approach is found in ref. [13].

In a spatially homogeneous neutral plasma with two ion species, it is convenient to describe collisional temperature relaxation[61] as

$$\frac{dT_\alpha}{dt} = -\nu_{\alpha\beta}(T_\alpha - T_\beta), \tag{9}$$

where $\alpha, \beta$ refer to different ion species. The collision frequency $\nu_{\alpha\beta}$ is a function of temperature, density, and charge. It can be generalized as

$$\nu_{\alpha\beta} = n_\beta \Phi \mathcal{S}, \tag{10}$$

where

$$\Phi = \left( \frac{Z_\alpha Z_\beta e^2}{4\pi\epsilon_0} \right)^2 \frac{\sqrt{m_\alpha m_\beta}}{(m_\alpha + m_\beta)^{3/2}} \left( \frac{1}{k_B T_{avg}} \right)^{3/2}, \tag{11}$$

and

$$T_{avg} = \frac{m_\alpha T_\beta + m_\beta T_\alpha}{m_\alpha + m_\beta}, \tag{12}$$

and $\mathcal{S}$ is a model-dependent collisional integral.

We have formulated three theoretical models with increasing fidelity to reveal their physics sensitivities in the plasma regime of our experimental and simulation results. In particular, our plasmas are strongly coupled and have mass ratios much closer to unity than the electron–ion mass ratio.

*Model 1.* Our first model for $\mathcal{S}$ in Eq. (10) is based on the well-known NRL Plasma Formulary result [see pp. 33–34 of ref. [62]].

The NRL model is obtained from the Fokker–Planck equation that contains the well-known Coulomb Logarithm (CL). In our plasmas, the NRL formulation for $\mathcal{S}$ is negative and cannot be used directly, indicating that strong scattering and screening is present, obviating the use of straight-line trajectories and standard Debye–Hückel screening models. To account for stronger scattering events, we extend the NRL straight-line-trajectory approximation to hyperbolic trajectories[28], here using cutoffs as in NRL,

$$\mathcal{S}^{(1)} = \frac{1}{2} \ln \left[ 1 + \left( \frac{b_{max}}{b_{min}} \right)^2 \right], \tag{13}$$

where

$$b_{max} = \left( \frac{1}{\lambda_1^2} + \frac{1}{\lambda_2^2} \right)^{-1/2}, \tag{14}$$

$$b_{min} = \left( \frac{Z_1 Z_2 e^2}{4\pi\epsilon_0} \right) \frac{1}{k_B T_{avg}}, \tag{15}$$

and $\lambda_\alpha^2 = \epsilon_0 k_B T_\alpha / (n_\alpha (Z_\alpha e)^2)$ is the Debye length of species $\alpha$. Eq. (13) always yields a positive definite result and gives the NRL result in the limit of large values.

*Model 2.* Our second model addresses strong correlations by modifying the screening length to be consistent with the effective screening length of an ionic transport model[15]. In contrast to the screening length in Eq. (14), we consider an alternate screening length that includes electron screening and a correction for strong ion coupling. Choosing $b_{max}$ to have the form

$$\lambda_{eff} = \left[ \frac{1}{\lambda_{TF}^2} + \sum_{\alpha=1}^{2} \frac{1}{\lambda_\alpha^2 + a_{ws}^2/x_\alpha} \right]^{-1/2}, \tag{16}$$

guarantees that the screening length does not vanish at low temperature, but rather approaches the (species dependent) interparticle spacing through the factor $a_{ws}^2/x_\alpha$ (see Methods). The choice of this functional form that includes $a_{ws}$ guarantees that the implied functional form of the binary interaction is consistent with numerical results in the strongly coupled regime[15]. The parameter $\lambda_{TF}$ is the Thomas-Fermi length calculated from the electron temperature, $T_e$, and density, $n_e$, see Eq. (23) in ref. [15] and represents electron screening. Note that Eq. (16) gives a positive definite value even at zero temperature. Our second model is then

$$\nu_{ij}^{(2)} = n_j \Phi \mathcal{S}^{(2)}, \tag{17}$$

$$\mathcal{S}^{(2)} = \frac{1}{2} \ln \left[ 1 + \left( \frac{2\lambda_{eff}}{b_{min}} \right)^2 \right]. \tag{18}$$

This model, when compared with the first, reveals the importance of strong interparticle correlations through Eq. (16).

*Model 3.* Our third model arises from the Chapman–Enskog expansion of the Boltzmann equation using an effective screened interaction to numerically obtain a cross section[15]. As such, consistent trajectories are included. There is no limit on the strength of scattering and cutoffs are not needed.

Through a Bhatnagar–Gross–Krook approach[61], the relevant collisional frequency can be identified to be

$$\nu_{ij}^{(3)} = n_j \Phi \mathcal{S}^{(3)}, \tag{19}$$

$$\mathcal{S}^{(3)} = \frac{128}{3} \frac{\sqrt{\pi}}{2^{3/2}} \mathcal{K}_{11}(g), \tag{20}$$

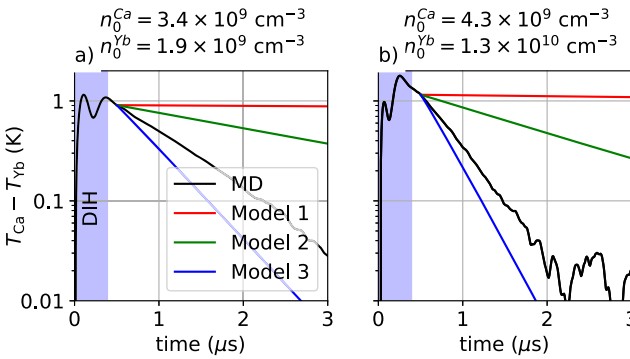

**Fig. 4 Temperature differences vs. time.** Semi-logarithmic plots of the temperature difference $T_{Ca} - T_{Yb}$ (black line) compared with temperature relaxation models are shown. Model 1 red line, Eqs. (10)–(15), Model 2 green line, Eqs. (17) and (18), Model 3 blue line, Eqs. (19) and (20). Simulation parameters $T_e = 100$ K, **a** $\kappa = 0.38$, **b** $\kappa = 0.46$. The systematic improvement of the models compared to the MD simulation is clearly shown. Data in **b** is reproduced from L. G. Silvestri, et al., Phys. Plasmas 28, 062302 (2021), with the permission of AIP Publishing.

where $g$ is the thermally-averaged ion-ion Coulomb coupling factor

$$g = \left(\frac{Z_1 Z_2 e^2}{4\pi\epsilon_0}\right)\frac{1}{k_B T_{avg}}\frac{1}{\lambda_{eff}}. \tag{21}$$

Typical values in our experiments and MD simulations are $g > 1$ (see Fig. 2). The collision integral $\mathcal{K}_{11}(g)$ is calculated from

$$\mathcal{K}_{11}(g) = \begin{cases} -\frac{1}{4}\ln\left(\sum_k^5 a_k g^k\right) & g < 1 \\ \frac{b_0 + b_1\ln g + b_2\ln^2 g}{1 + b_3 g + b_4 g} & g > 1 \end{cases} \tag{22}$$

where $a_1 = 1.4660$, $a_2 = -1.7836$, $a_3 = 1.4313$, $a_4 = -0.55833$, $a_5 = 0.061162$, $b_0 = 0.081033$, $b_1 = -0.091336$, $b_2 = 0.051760$, $b_3 = -0.50026$, $b_4 = 0.17044$.

**Comparison of the experiment, simulation, and theory.** In Fig. 3e, f, we plot the Ca$^+$ and Yb$^+$ ion temperatures differences from both the experiment and the MD simulations. The temperature differences show excellent agreement between the laboratory measurements and MD simulations. Given the velocity flow locking shown in Fig. 1 and also the dramatic flattening of the velocity gradient, we conclude that the hydrodynamic expansion has a negligible effect on the temperature measurements and the temperature difference, especially for $t < 1.5$ µs.

In Fig. 4, we compare MD results with the theoretical predictions presented above. The three models can be compared with MD results only after each species has reached a Maxwellian velocity distribution. As presented above the two species can be considered to be Maxwellian after $t \sim 1$ µs. As stated previously, the progression of models begins with standard plasma theory that incorporates physically motivated ion trajectories, Eqs. (10)–(15)[28]. This is plotted as a red line and it indicates a serious deficiency of standard plasma theory. This is expected since this model, Eq. (13), uses the ion Debye length as the maximum length scale.

The next level of improvement, Model 2, provides a correction for strong coupling when the Debye length becomes unphysically small, Eqs. (17) and (18). This is plotted as a green line. This strong coupling correction brings the theory closer to the simulations, but still underpredicts the relaxation rates.

Our best model, Model 3, is based on an effective potential in a Boltzmann description, Eqs. (19)–(21), which includes velocity-dependent strong scattering. This is plotted as a blue line in Fig. 4. This incorporates strong scattering in a self-consistent way,

reducing the ambiguity in choosing ad hoc cut-off parameters inherent in a CL approach. Nonetheless, this model predicts temperature relaxation rates somewhat faster than the MD result. This is surprising given the previously demonstrated accuracy of this model in reproducing experimental data and MD simulations of momentum transfer, joule heating, diffusion, viscosity, thermal conductivity, etc.[9,15,21].

This overprediction may be due to errors in the effective potential, non-binary collisions, or coupled modes[33,34,63,64]. The influence of the coupled mode can be identified when examining the dielectric response function $\varepsilon(\mathbf{k}, \omega)$ of a binary plasma mixture. In the case of a binary ionic mixture, the temperature relaxation equation for species 1 is[13]

$$\frac{dT_1}{dt} = \iint \frac{dk\,d\omega}{3n_1\pi^3}\, k^2\left(\frac{U_{12}(\mathbf{k},\omega)}{|\varepsilon(\mathbf{k},\omega)|}\right)^2 \left\{ T_1\,\mathrm{Im}\left[\Pi_{21}A_{22}^*\right]\mathrm{Im}\,\hat{\chi}_{11}^{(0)}\right.$$
$$\left. - T_2\,\mathrm{Im}\left[\Pi_{12}^*A_{11}\right]\mathrm{Im}\,\hat{\chi}_{22}^{(0)}\right\}. \tag{23}$$

With exchange $1 \rightarrow 2$ we obtain the equation for species 2. In the above equation $U_{12}(\mathbf{k}, \omega)$ is the Fourier transform of the Yukawa interaction between the two ion species, $\hat{\chi}_\alpha^0(\mathbf{k}, \omega)$ is the external response function of species $\alpha$, $A_{\alpha\beta} = \delta_{\alpha\beta} - \tilde{U}_{\alpha\beta}\Pi_{\alpha\beta}(\mathbf{k}, \omega)$, and the elements of the matrix $\Pi(\mathbf{k}, \omega)$ are defined by

$$\Pi_{\sigma\sigma'}(\mathbf{k}, \omega) = \chi_\sigma^{(0)}(\mathbf{k}, \omega)[1 - G_{\sigma\sigma'}(\mathbf{k}, \omega)], \tag{24}$$

where $\chi_\sigma^{(0)}(\mathbf{k}, \omega)$ is the free particle polarizability and $G_{\sigma\sigma'}(\mathbf{k}\omega)$ is the local field correction. The three models described above neglect the frequency dependence of $\varepsilon(\mathbf{k}, \omega)$ and consider only the static version $U_{12}(k)/\varepsilon(\mathbf{k}, 0)$. This effective interaction is then used to inform the CL in a Fokker-Planck approach (Model 1 and Model 2) or the cross-section in a Boltzmann equation (Model 3). It is worth noting that Eq. (23), which describes the interaction of classical ions, has wide applicability to most non-ideal plasmas, with transferability guaranteed through the choice of the most appropriate pair interaction[65].

In general, the effective interaction is time and frequency-dependent. The electron–ion and ion–ion dynamics need to be considered when extending plasma models to include coupled modes. Some of these processes harden the ion–ion potential while others soften it. One trend, observed in ref. [13] for one particular density ratio, is that mode coupling becomes more significant as the ion mass ratio approaches unity. Future work is needed in this direction.

## Discussion

We demonstrate that dual-species UNPs provide a platform for studying ion transport properties in a two-temperature system. We characterized the approach to equilibrium using a velocity moment analysis, which reveals when a relaxing system can be safely treated as a "two-temperature" system. This powerful approach facilitates the comparison of data with quasi-equilibrium theories.

We present a measurement of ion-ion temperature relaxation rates in a strongly coupled binary ionic mixture. Using a single diagnostic method we directly measure the ion temperatures without inference through, for example, an equation of state. This shows the remarkable capability for UNPs to investigate component physics across coupling regimes, simulating some aspects of, e.g., HEDP, plasma mixtures, and liquid metal alloys.

We show that our MD simulations of temperature relaxation agree with experimental measurements. This reinforces the fact that the Yukawa potential, Eq. (5), accurately describes ion-ion interaction in dual-species UNP mixtures. This further confirms the ability of our MD simulations to capture a very complex

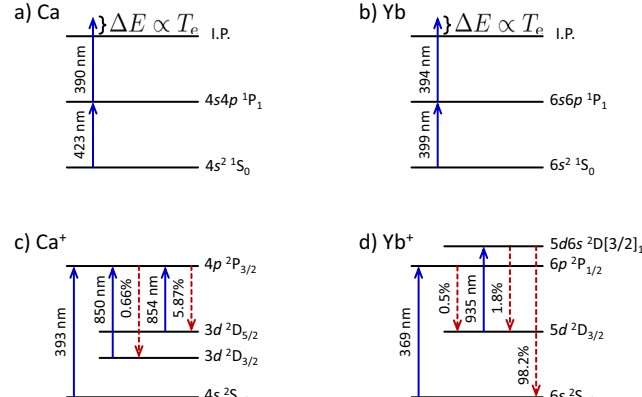

**Fig. 5 Atomic energy levels and laser wavelengths.** Blue arrows in **a** and **b** indicate the lasers used for ionizing Ca and Yb, respectively. The electron energy is given by the difference between the ionization limit and the laser photon energy. **c**, **d** show levels used in Ca+ and Yb+ spectroscopy, respectively. The blue arrows indicate the laser-driven transitions while the red dashed arrows indicate the spontaneous emission with the associated branching fractions. In this experiment, we use a Ti:sapphire laser to repump the 854 nm transition in Ca+. Repumping the transition at 850 nm does not change the measured results. We do not repump the 935 nm transition in Yb+.

relaxation process. This confidence, in turn, allows us to employ the MD as a surrogate for information that the experiment cannot provide. These observations reinforce the importance of having MD as an integral part of experimental workflows.

We compare the simulated relaxation rates with three popular temperature relaxation theories of varying fidelity. The closest theory is based on solving the Boltzmann equation using an effective potential. The variance between this theory and the MD simulations is likely caused by coupled modes, an effect that is omitted from the theory by design. Future work could explore the influence of coupled modes on ion transport. Incorporating coupled modes into the Boltzmann solutions could also prove fruitful for ion transport in the regime of relatively small mass ratios.

## Methods

**Experimental details.** We use resonant two-stage photo-ionization to ionize 100% of Ca atoms and up to 60% of Yb atoms. The wavelengths are shown in Fig. 5. The ionization pulses are offset by 40 ns to prevent the Ca ionization pulses from ionizing the Yb atoms. Otherwise, cross-ionization between the two plasmas would confound the electron temperature, as shown in Fig. 6. The electron temperature is determined by the photon energy of the ionizing pulse above the ionization threshold, and $T_e = 96$ K in these experiments. The initial plasma densities are controlled by expanding the neutral atom cloud prior to ionization, also illustrated in Fig. 6. The Ca and Yb atom clouds expand for different lengths of time up to 2 ms, allowing independent control of the relative densities of each species. The process of loading the neutral atom trap, expanding the neutral atom clouds, and generating the plasma takes several ms and the process is repeated at a rate of 10 Hz.

The Ca+ and Yb+ velocity distributions are measured using laser-induced fluorescence. Probe laser beams at 393 (Ca+) and 369 nm (Yb+) are overlapped using a dichroic mirror and then coupled into a single-mode polarization-maintaining optical fiber. The fiber output is collimated with a Gaussian waist of 3.9 mm and then cylindrically focused ($f_{cyl} = 350$ mm) to an rms thickness of 0.15 mm to illuminate a sheet of ions in the center of the plasma[56] (see Fig. 7). The probe laser beam intensities are typically 10–20% of the saturation intensity.

For long interrogation times, $t > 1$ μs, optical pumping is potentially problematic. For the 393 nm transition in Ca+, lasers at 854 and 850 nm could be used to prevent optical pumping into the metastable $^2D$ states[66]. Throughout these experiments, we use a laser at 854 nm to prevent optical decay into the $3d^2D_{5/2}$ level. We initially also used a laser at 850 nm to prevent optical decay into the $3d^2D_{3/2}$ level. However, we found that including this laser did not change the measurement results. In later experiments, the 850 nm laser was not used.

We further verify that optical pumping is a negligible source of error in both Ca+ and Yb+ by turning the probe laser beams on for short periods of time and delaying the turn-on time for up to 10 μs. We find that for our probe laser

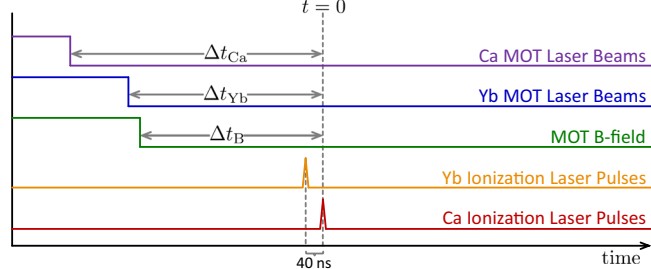

**Fig. 6 Timing diagram for density selection and plasma creation.** A step up indicates turning on and a step down indicates turning off the indicated quantity. The Ca and Yb neutral atom clouds are expanded for $\Delta t_{Ca}$ and $\Delta t_{Yb}$ to obtain the desired densities, typical expansion times range between 0.5 and 2 ms. The pulse delay for the ionization pulses is 40 ns to eliminate cross ionization to preserve the known electron temperature of the plasma. Otherwise, the Ca+ ionization pulses would also ionize the Yb atoms and produce a very high electron temperature. Due to the large ion mass, the Yb+ plasma does not expand on this 40 ns time scale. Data plotted in previous figures also have this 40 ns delay. Its influence is imperceptible in the data. The MOT magnetic field is turned off $\Delta t_B = 500$ μs prior to ionization.

intensities, there is no difference between these measurements and those derived from leaving the probe lasers on all of the time.

Details of the laser-induced fluorescence measurement process are given in Fig. 7. With the probe laser frequencies at a particular offset from resonance, we collect fluorescence as a function of time after the plasma is created. Fluorescence measurements from up to 100 identical plasmas are averaged at a given frequency of the probe lasers. The probe laser frequency is changed and the measurement process is repeated. Using 11 different probe laser frequency offsets, we sample the ion velocity distributions, using the Doppler shift to convert frequency offset to ion velocity. The data is post-processed so that at a given time after ionization, the fluorescence signal as a function of frequency is fit to a Voigt profile, and the rms Gaussian width, $v_{\alpha,th}$, is used as a fit parameter. This is used to determine the ion temperature, $T_\alpha$. Using the ICCD camera allows us to collect fluorescence averaged over a particular time. Typical averaging times are 50 to 500 ns, usually 10% of the delay time, depending on the ion dynamics under consideration.

**Derivation of the fluid expansion model.** We model the UNP as a three-species plasma composed of electrons and two ionic species. The species momentum equations are obtained from the velocity moments of the underlying kinetic description and are given by

$$n_\alpha \frac{\partial \mathbf{u}_\alpha}{\partial t} + n_\alpha \mathbf{u}_\alpha \cdot \nabla \mathbf{u}_\alpha + \frac{1}{m_\alpha} \nabla p_\alpha - \frac{\mathbf{F}_\alpha}{m_\alpha} n_\alpha = \mathcal{C}_\alpha, \qquad (25)$$

where $\alpha = \{e, Ca, Yb\}$. Apart from the time derivative, this model describes advection, pressure forces, external and internal forces, and collisions. Approximations relevant to our ultracold plasma experimental conditions can be made, and include steady-state electrons ($\partial \mathbf{u}_e / \partial t \approx 0$ on the ion time scales of interest here), small velocities among all species ($\mathbf{u}_\alpha \cdot \nabla \mathbf{u}_\alpha \approx 0$), vanishing pressure for the (ultracold) ions ($p_{Ca}, p_{Yb} \approx 0$), internal electrostatic forces and a friction term between ionic species. With these approximations, Eq. (25) become

$$\nabla P_e = n_e \mathbf{F}_e, \qquad (26)$$

$$\frac{\partial \mathbf{u}_1}{\partial t} = \frac{\mathbf{F}_1}{m_1} - \nu_{12}^m (\mathbf{u}_1 - \mathbf{u}_2), \qquad (27)$$

$$\frac{\partial \mathbf{u}_2}{\partial t} = \frac{\mathbf{F}_2}{m_2} - \nu_{21}^m (\mathbf{u}_2 - \mathbf{u}_1), \qquad (28)$$

where the subscripts $1 = $ Ca and $2 = $ Yb. These equations are coupled to the species continuity equations and a Poisson equation for the electric fields. Assuming the electrons are isothermal and $Z = 1$, we obtain

$$\frac{\partial \mathbf{u}_1}{\partial t} = -\frac{k_B T_e}{m_1} \frac{\nabla n_e}{n_e} - \nu_{12}^m (\mathbf{u}_1 - \mathbf{u}_2), \qquad (29)$$

$$\frac{\partial \mathbf{u}_2}{\partial t} = -\frac{k_B T_e}{m_2} \frac{\nabla n_e}{n_e} - \nu_{21}^m (\mathbf{u}_2 - \mathbf{u}_1). \qquad (30)$$

**Simulation details.** MD simulations are performed using methods described in ref. [13]. For each experimental condition, five non-equilibrium simulations with

**Fig. 7 Experimental details for fluorescence detection. a** Schematic diagram of the optical system used for fluorescence detection. The plasma is illuminated by cylindrically focused laser light. Laser-induced fluorescence is collected using an $f/2$ 1:1 imaging system. For PMT measurements the plasma is imaged onto a $\phi = 250\ \mu m$ aperture, enabling measurements of the central portion of the plasma. Spectral filters in front of the PMTs allow simultaneous measurements of fluorescence from both $Ca^+$ and $Yb^+$ ions. A spectral filter is also used before the ICCD camera. **b** Typical $Ca^+$ fluorescence ICCD camera image, integrated over probe laser frequency, when $Yb^+$ ions are present after a time-evolution of $5\mu s$. Camera measurements give spatial information at a particular time after ionization.

different initial conditions were performed on an Intel Core i7-8700K and 48 GB of RAM. The typical runtime for a single simulation run was ~20 h.

The experiment uses a 40 ns delay between the Yb and Ca ionization steps. We performed more simulations without the initial time delay of 40 ns and found no difference in the results. A back-of-the-envelope calculation shows that the averaged displacement of the $Yb^+$ ions in the first 40 ns is ~100 nm, hence, not a significant change over the length scales of the system.

**Comparison of experiment and simulations**. To enforce compatibility in comparing the MD data to the experiment, the MD velocity distribution is convolved with a Lorentzian distribution and then fit a Voigt profile with the Gaussian width as a fit parameter. This slightly underestimates the average ion kinetic energy during the DIH phase because of slight departures from a Maxwellian velocity distribution. Experimentally, we have access to the ion velocity distribution through the fluorescence signal. This signal is necessarily a convolution of the velocity distribution with the natural lineshape of the atomic transition. The half-width at half-maximum of the (Lorentzian) $Ca^+$ 393 nm transition is 11.5 MHz[67]. This corresponds to a velocity of 4.5 m/s. This is small compared to the post-DIH (Gaussian) rms velocity of 20 m/s for $Ca^+$ ions at a temperature of 2 K. Convolving the MD velocity distribution and fitting to a Voigt profile reduces the ion temperature by a few percent during the DIH phase, as illustrated in Fig. 5 of ref. [56].

**Theoretical details**. The models consider a spatially homogeneous plasma composed of two ion species with different masses, $m_\alpha$, different number densities, $n_\alpha$, and charge numbers, $Z_\alpha$. The surrounding negative electronic background is at temperature $T_e$ and density $n_e = Z_1 n_1 + Z_2 n_2$. The Wigner–Seitz radius is defined from the total ion number density, $a_{ws}^3 = 3/(4\pi n_{tot})$, $n_{tot} = n_1 + n_2$. The concentration of each ion species is $x_\alpha = n_\alpha/n_{tot}$. The electron density $n_e = n_{tot}$ when $Z_1 = Z_2 = 1$, as is true for our UNPs. The ion plasma frequency of species $\alpha$ is $\omega_\alpha^2 = (Z_\alpha e)^2 n_\alpha/(\epsilon_0 m_\alpha)$. The total ion plasma frequency is $\omega_p^2 = \sum_\alpha \omega_\alpha^2$. The ion thermal speed of species $\alpha$ is given by $v_{\alpha,th} = \sqrt{k_B T_\alpha/m_\alpha}$. Typical equilibrium values in our experiments are $\Gamma_\alpha = 3$, as shown in Fig. 2.

## Data availability

The data generated in this study are deposited in the Zenodo database, https://zenodo.org/record/5648297[68].

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

## Acknowledgements

We thank Dr. Jeffrey Haack of Los Alamos National Laboratory for useful conversations. M.S.M. and L.G.S. were supported by the U.S. Air Force Office of Scientific Research Grant no. FA9550-17-1-0394. R.T.S. and S.D.B. acknowledge support from the U.S. Air Force Office of Scientific Research Grant no. FA9550-17-1-0302 and the National Science Foundation Grant no. PHY-2009999.

## Author contributions

The authors contributed equally to this work. R.T.S. and S.D.B. Conceived and built the experiment, designed and carried out the laboratory measurements, and performed the experimental data reduction. L.G.S. and M.S.M. designed and performed the computer simulations and theoretical analysis and calculations. All authors contributed equally to the writing of the paper. All authors commented on the paper and agreed on its contents.

## Competing interests

The authors declare no competing interests.
