## [Peer Review File · Nature Communications]

Temperature relaxation in strongly-coupled binary ionic mixturesREVIEWER COMMENTS

Reviewer #1 (Remarks to the Author):

Report on Nature_Communication_295451/Sprenkle et al.

In the manuscript by Sprenkle et al (submitted to Nature Communications), the authors presented both experiments and MD calculations (+ model predictions) of ion-ion temperature relaxation in ultracold neutral plasmas (UNPs) of dual species [Ca+ & Yb+]. The table-top UNP experiments were well designed and carried out nicely, in which the temperature history of each species is inferred by measuring laser induced fluoresce. The classical molecular dynamics (MD) simulations, with Yukawa type potentials, reproduced the temperature history for relatively low-density ($\sim 10^9/\text{cc}$) UNPs, but failed to predict the absolute temperature variation with time for high-density case ($\sim 10^{10}/\text{cc}$ for Yb+) [Fig. 3b]. The authors further discussed how the three temperature relaxation models are unable to predict the ion-ion temperature relaxation rate at both plasma conditions.

Given the demonstrated experimental capability of using UNPs to probe important plasma physics at the strongly coupled regime, this work should be justified for its publication in the high-impact journal of Nature Communications, provided that the authors are willing to improve the current version of their manuscript. In case the authors choose this path to publish their work, I would like to give the following suggestions (even so some are minor):

1. To be frank, I was lost to read through the Introduction that is full of scattered discussions of HEDP and ICF topics of which most are even irrelevant to the central topic of the current work. I would love to see the authors focus on: (a) why is an experimentally benchmarked ion-ion temperature equilibrium model important to HEDP/ICF in the strong coupling regime? (b) How a precisely controlled UNP experiment can provide a useful platform for tackling nasty problems in HEDP? And (c) what influence of such UNP experiment might shed light on transport model construction for strongly coupled plasmas?
2. It does not hurt to giev the full expression of the function $K_{11}(g)$ in Eq. (13); Nobody likes to trace to another paper to find them out (As far as I know, there is no page limit for Nature Communications).
3. On page 9, the authors mentioned that they shut-off Yb+ interactions with Ca atoms for the first ~ 40 ns in their MD simulations (by following experiments). It's not clear why experiments need to create the two-species UNPs with a time delay; and how the non-adiabatically switching on of Ca+ at $t=40$ -ns in MD will affect their simulation results. Could this latter process affect more the dense Yb+ case so that large discrepancies were seen in Fig. 3(b)?

4. The authors need a thorough discussion on the possible reason(s) that the MD simulations failed to reproduce the absolute temperature history for the high-density case [Fig. 3(b) again]. For now, I would view the agreement of temperature difference between MD and experiment [Fig. 3(d)] to be just “accidental”. Without reproducing the experimentally-measured temperature history of both species at high densities, I would not be confident to say “...the Yukawa potential, Eq. (17), accurately describes ion-ion interaction in dual-species UNP mixtures” that the authors stated on page 13.

5. It would be beneficial for the coupled-plasma physics community to see how these results will help to build accurate transport physics models for HED plasmas that are many orders of magnitude dense than UNPs. In other words, the discussions on page 12 need to be more concise.

Reviewed by S. X. Hu @ Laboratory for Laser Energetics, University of Rochester

Reviewer #2 (Remarks to the Author):

In the manuscript "Temperature relaxation in strongly-coupled binary ionic mixtures" by R. Tucker Sprenkle et al. the authors report on experimentally measured velocity profiles obtained by laser-induced fluorescence of ions in an ultracold plasma after ionizing laser-cooled atoms in a dual species magneto-optical trap (MOT). The inferred temperature evolution over time is compared to molecular dynamics (MD) simulations as well as various relaxation models.

The results in general are original and add new insights connected to a broad range of plasma systems beyond ultracold plasma. The work addresses urgent questions in transport properties of strongly coupled Coulomb systems. Moreover, the experimental approach as well as the theoretical treatment are state-of-the-art, and the results are, in general, well communicated. However, before publication I would prefer having (potentially easy to obtain) additional experimental data on the spatial evolution of the dual species plasma. Additionally, the structure of the manuscript must be rearranged for a clearer presentation. Finally, I am missing a few technical details and have more detailed questions and remarks below. For the above reasons I recommend a major revision of the manuscript before publication.

The reader of the manuscript first needs to go through detailed theoretical models before it is clear what the experiment is accessing. I recommend restructuring the manuscript so that the experimental observation is presented first and then compared to MD simulations, which are compared to various theoretical modes. Moreover, the manuscript does not make use of the usual "Methods" section in Nature Communications for outsourcing detailed methodological details. I strongly recommend moving experimental and theoretical details there and refer to it when the essential ideas / outcomes and the gist of the models and measurements are discussed in the main manuscript.

Apart from the suggested general improvements above, I have more specific questions and comments on the content of the manuscript:

1. Paragraph three of the Introduction (p. 2) discusses the validity of transport models and refers to limits of the theory as well as the necessity of using hybrid models. I wonder when the limits of the discussed models set in in terms of typical density / temperature regimes of a plasma?
2. In the third paragraph on p. 3 it is discussed that the physics in low density / low temperature systems can be mapped to high-energy-density plasma (HEDP). A short comment on why the mapping is possible and how it works would be good for the broader audience of Nature Communications.
3. The authors claim a coupling parameter of $\Gamma = 3$ in equilibrium. A short comment on how the dynamics / temperature relaxation in a strongly coupled plasma is expected to be different from a non-strongly coupled plasma would be appreciated. Moreover, is strong-coupling necessary for HEDP simulators (as claimed in paragraph four on p. 3)?
4. I am missing a few references to recent experiments on the evolution of ultracold plasma in the strong-coupling regime such as [T. K. Langin et al., Science 363, 61 (2019)], [T. Kroker et al., Nat. Commun. 12, 596 (2021)], [M. A. Viray et al., Phys. Rev. A 102, 033303 (2020)] and [E. V. Crockett et al., Phys. Rev. A 98, 043431 (2018)] which the authors might want to consider.
5. In the last paragraph of the introduction (p. 4) it is discussed that classical MD simulations are sufficient, however, earlier (p. 3) the authors point out that plasma models "must include effects of Fermi degeneracy [...]". I am curious under which parameters Fermi degeneracy is expected to be of relevance, especially related to the experiment of the authors.
6. What do the authors mean by "same sign of charge" in the last paragraph of the introduction (p. 4)? Simply that the plasma consists of an ion-ion mixture? Why are the electrons (with negative charge) not relevant then?

7. The "Theoretical Considerations" (p. 4) assume a spatially homogeneous plasma while most experiments (including the conducted experiment) are limited to Gaussian shaped plasma. I would appreciate a short comment on the expected differences and why the authors consider a homogeneous plasma in the modelling.

8. To remove ambiguities, $a_{ws}^3 = 3 / 4 \pi n_{tot}$ should be formatted as $a_{ws}^3 = 3 / (4 \pi n_{tot})$ on p. 4.

9. On p. 4 the authors refer to Γ as the "strong coupling plasma parameter". I'd rather call it coupling parameter because it distinguishes between strongly and weakly coupled plasma and is valid for both regimes.

10. The authors refer to a coupling parameter of $\Gamma = 3$. For clarity, it would be good to explicitly state which typical densities and ion temperatures this relates to in the experiment.

11. On p. 5 the authors state that the "first model is found in the NRL Plasma Formulary". Please explicitly reference the model for clarity. I assume the authors mean Eq. (3-5) with the first model.

12. On p. 5 the authors refer to p. 33 in the NRL Plasma Formulary (2013). I find the relevant formulas one page later on p. 34 in the 2013 NRL Formulary. Additionally, I am confused about the units. The cross section in Eq. (5) does not seem to have the unit $[m^2]$. The collision frequency in Eq. (4) should have the unit of a rate $[1/s]$ but combining a cross-section Φ in Eq. (5) $[m^2]$ and number density $[1/m^3]$ with the collision integral in form of the Debye length $[m]$ results in a unitless quantity. Even more confusing is the fact that the collision integrals for the other models in Eqs. (6,11,13) are dimensionless (unlike to $S = \lambda_{\alpha} \lambda_{\beta}$ having the unit $[m]$).

13. Some physical sense of a negative Debye length (p. 5) or why it appears would be helpful.

14. Eqs. (7-8) seem to differ from the treatment on hyperbolic trajectories in Ref. (25). Can the authors elaborate which model in Table I in Ref. (25) they refer to and how they arrive at the min. / max. distances in Eq. (7-8)? Moreover, are λ_1 and λ_2 equal to λ_{α} and λ_{β} ?

15. On p. 6 the authors state that by using Ref. (25) the NRL model is generalized to ion-ion collisions. Does this mean they are not included at all in the NRL approach?

16. Is x_α in Eq. (9) the concentration n_α/n_{tot} ? And is Eq. (9) derived from Eq. (35) in Ref (18)? If so, can the authors briefly elaborate how they arrive at the result or if further assumptions / approximations are made? Again, in Eq. (9) the species are not referred to as α and β but 1 and 2 with α as dummy index in the sum. That must be consistent in the manuscript.

17. What do n and m in $n = m = 1$ refer to on the bottom of p. 6?

18. In Fig. 1(a) the experimental setup is presented. While there are filters on the PMT branch, the light to the ICCD camera does not seem to be filtered. However, Fig. 1d presents Ca⁺ fluorescence selectively. Is this achieved by solely shining in the Ca⁺ fluorescence laser? The 250 μm pinhole sets a boundary for the extent of the probed central part, however, I am missing the width of the cylindrically focused laser light along the imaging axis to get an idea of the area the signal is integrated over. Is the typical PMT signal in Fig. 1(b-c) for Ca⁺ or Yb⁺ ions? I am missing a discussion of Fig. 1(d). The subfigure needs a scale bar to get an idea of the size of the plasma. Moreover, I am curious about the spatial structure visible in the image. Why is there a "wing" left to the Gaussian spot in the center? As the authors seem to be able to take snapshots of the spatial evolution of the central dual species plasma region in parallel to the PMT signals, I would be happy to see a short series of images on how the plasma evolves in real space and if this expansion is consistent with the measured velocity profiles.

19. Is the peak density in the MOT n_0 on p. 7 referring to the total atomic density or the peak density for each species?

20. I suggest including a reduced overview level scheme of Ca and Yb atoms and all involved wavelengths (MOT and photoionization), which would make following the experimental procedure a lot easier. The same holds for the Ca⁺ and Yb⁺ ions involving all probe and optical pumping lasers.

21. I am missing the wavelengths used for photoionization in the manuscript. Moreover, does an excited state fraction in the MOT alter the photoionization scheme (e.g. different electron temperatures resulting from a non-resonant two-photon ionization process)? If so, do the authors know the excited state fraction in the MOT?

22. The authors state that the electron temperature is around 96 K. Does this mean the electron temperature is experimentally tunable? If so, in what range? And does a modified electron temperature change the initial ion temperature?

23. Are the initial plasma densities independently tunable for each species in the experiment?
24. What is the pulse duration of the ionization laser pulse? What is the duration of the probe pulse?
25. Do the authors know how sensitive the experiment / expansion dynamics is to electric stray fields and to what degree are external fields under control?
26. On p. 7 it is stated that the authors "verify that optical pumping is a negligible source of error". A brief sentence how this is achieved would be appreciated.
27. On p. 8 the authors discuss that disorder induced heating (DIH) increases the temperature to near 1 K. How does a flat atom pair distribution function relate to a significant Coulomb coupling of $\Gamma = 3$ (p. 4) where the ions are expected to form spatial correlations?
28. What is a typical runtime (minutes / hours / weeks) of the MD simulation and what computer / cluster has been used to carry out the simulation?
29. On p. 8 it is stated that the Yb ions are created first, and Ca is ionized 40 ns later. Why is the ionization not performed simultaneously? In general, a sketch of the experimental sequence would be very helpful. How long is the MOT phase, ionization sequence etc.?
30. Please explicitly state the experimental electron temperatures and densities used for deriving the value of κ on p. 9 for clarity.
31. Am I correct that the MD simulation calculates the dynamics of the $N = 50,000$ ions (p. 10) but not explicitly the trajectories of the involved electrons?
32. On p. 11 it is stated that the MD velocity distribution is convolved with a Lorentzian distribution. What is the width of the Lorentzian distribution, and does it relate to the experimental linewidth of the laser? Connected to this: Please state the experimental velocity resolution resulting from the laser linewidth. The unconvolved simulation result would show the more fundamental dynamics of the temperature time-evolution. The authors might consider showing the raw results of the simulation in an inset or figure in the Supplementary Material.

33. What is the origin of the oscillations in temperature within the first 500 ns in Fig. 3 and is the oscillation frequency connected to the plasma frequency?

34. On p. 11 the authors mention the influence of the density gradient in the probed region. What is the width of the light sheet (i.e. the probed region) and the density modulation over this region for both experimental regimes? Why is the density and size of the Yb atoms modified stronger (one order of magnitude for the density) compared to the Ca atoms ($\sim 20\%$) in Fig. 3?

35. On p. 12 the authors mention non-binary collisions as reason for the overprediction. I am wondering what the influence of three-body recombination (TBR) into Rydberg states is. Can the authors estimate the TBR rate for the dual species plasma?

36. I am missing a few details in Fig. 4. The comparison to the theoretical models in Fig. 4 seems to start after $1 \mu\text{s}$ or do all models fall onto the MD curve for shorter times? Why are the models not compared to the MD data for shorter evolution times? Which equation is used for the experimental fit from which the parameter $\nu = 2.52 \text{ MHz}$ is extracted? Can we deduce a cross section or collision integral from that?

37. The authors claim a strongly coupled binary ionic mixture. The coupling parameter is given as $\Gamma = 3$, however, depends on the time-evolution of the density and temperature. Can the authors estimate how Γ evolves with time in the experimental regimes?

Reviewer #3 (Remarks to the Author):

Apparently this text is the first in reporting a detailed and quantitative assessment for the ion-ion energy relaxation (ER) in a binary ionic plasma involving an experimental protocol supplemented by a MD simulation based on a Yukawa ion-ion effective potential. The authors claim that elaborating through the UNP plasma specificities allow them to perform a quantitative experience-simulation meaningful match while securing transferability of their conclusions to more mundane plasmas: ICF- or WDM-like.... This point should be strengthened more convincingly.

Moreover, the three models confronted on pp. 5-6 should be given more details about their physics content. It is not sufficient to drive the reader to the third author former theory works.

Editing remarks:

-p.5 line 8/bottom....Lamda with suffixes alpha beta (CL) should be explained

-p.6 line 7/top....Lamda TF needs also to be detailed in view of its significance

-p.14 Ref.14....-Taylor,Richtmyer-Meshkov....

-p.20 Ref.53....BGK model...

-p.20 Ref.55....Yukawa

-p.26 Caption Fig.3 apparently Sigma Ca and Sigma Yb are not introduced neither in text nor here

Reacting to these suggestions could allow the text to be considered for publication in Nat-Com.

NCOMMS-21-03617: Response to Referees

Reviewer 1

We thank the reviewer for insightful comments. We excerpt direct requests in black and indicate our response in red.

1. (a) why is an experimentally benchmarked ion-ion temperature equilibrium model important to HEDP/ICF in the strong coupling regime?

The beginning of this answer is in the paragraph starting in line 36. In the next paragraph, starting at line 47, we discuss the complications with existing temperature equilibration work. The following paragraph, starting in line 57, discusses the two-temperature problem at a fundamental level. Finally, in the paragraph starting on line 67 the connection between HEDP and UNPs is explicitly addressed.

- (b) How a precisely controlled UNP experiment can provide a useful platform for tackling nasty problems in HEDP?

See previous comment

And (c) what influence of such UNP experiment might shed light on transport model construction for strongly coupled plasmas?

Part of our response to this question is mentioned above. Our discussion continues in the paragraph starting on line 276 and Eqs (21) and (22).

2. It does not hurt to give the full expression of the function $K_{11}(g)$ in Eq. (13).

We added the equation for $K_{11}(g)$ in Eq. (20).

3. On page 9, the authors mentioned that they shut-off Yb⁺ interactions with Ca atoms for the first <40 ns in their MD simulations (by following experiments). It's not clear why experiments need to create the two-species UNPs with a time delay

Experimentally, the 390 nm laser pulse used to ionize Ca will also ionize any Yb atoms in the excited state. The 40 ns delay is used to ensure that no Yb atoms are ionized by the Ca ionization pulses. Any "cross-ionization" by the wrong laser pulses would result in a very high and uncontrolled electron temperature. This is explained in line 324 in the manuscript.

and how the non-adiabatically switching on of Ca⁺ at t=40-ns in MD will affect their simulation results. Could this latter process affect more the dense Yb⁺ case so that large discrepancies were seen in Fig. 3(b)?

The adiabatic switching on the Ca⁺ ions at 40 ns doesn't influence the equilibrium temperature in the MD. We verified this by turning on the Ca⁺ ions at 0 ns and found no change in the equilibrium values. We explain this starting in line 396.

Starting in line 253 we discuss the experiment/MD discrepancy. On the experimental side, we modeled the effects of the camera point spread function and the focusing geometry of the probe laser beam. Neither effect explains the discrepancy between MD and theory. A paper by Killian [PoP 22, 033513 (2015)] saw a similar amount of heating and attributed it to ion acoustic wave heating. Because the DIH temperature is related to the density, any locally high density variations will have a somewhat higher DIH temperature. As this local temperature variation relaxes (a very fast process) the equilibrium temperature increases.

4. The authors need a thorough discussion on the possible reason(s) that the MD simulations failed to reproduce the absolute temperature history for the high-density case [Fig. 3(b) again]. For now, I would view the agreement of temperature difference between MD and experiment [Fig. 3(d)] to be just "accidental". Without reproducing the experimentally-measured temperature history of both species at high densities, I would not be confident to say "...the Yukawa potential, Eq. (17), accurately describes ion-ion interaction in dual-species UNP mixtures" that the authors stated on page 13.

The ion acoustic wave heating could account for the discrepancy. It would allow the temperatures to be higher but the temperature difference to match the MD results. IAW heating is smaller when the neutral atom clouds expand for longer times, which is the case for the low-density plasmas in Fig 4a,c.

5. It would be beneficial for the coupled-plasma physics community to see how these results will help to build accurate transport physics models for HED plasmas that are many orders of magnitude dense than UNPs. In other words, the discussions on page 12 need to be more concise.

We expanded the discussion slightly starting in line 276 and surrounding Eq. (21).

Reviewer i Green

We thank the reviewer for the considerable effort devoted to reviewing our manuscript. We excerpt direct requests in black and indicate our response in green.

- I would prefer having additional experimental data on the spatial evolution of the dual species plasma.

We have added a new Fig. 1 that shows the spatial evolution of the plasma. The discussion of that evolution, starting at line 109, includes measurements of the velocity profile and a justification for modeling the central portion of the (Gaussian) UNP as a uniform density plasma.

- The structure of the manuscript must be rearranged for a clearer presentation ... I recommend restructuring the manuscript so that the experimental observation is presented first and then compared to MD simulations, which are compared to various theoretical models...

We have extensively re-written the manuscript to follow these recommendations.

1. Paragraph three of the Introduction (p. 2) discusses the validity of transport models and refers to limits of the theory as well as the necessity of using hybrid models. I wonder when the limits of the discussed models set in in terms of typical density / temperature regimes of a plasma?

We added a discussion of the Coulomb coupling parameter and noted failures when $\Gamma > 1$. See discussion starting on line 36.

2. In the third paragraph on p. 3 it is discussed that the physics in low density / low temperature systems can be mapped to high-energy-density plasma (HEDP). A short comment on why the mapping is possible and how it works would be good for the broader audience of Nature Communications.

We added that discussion in line 67.

3. The authors claim a coupling parameter of $\Gamma = 3$ in equilibrium. A short comment on how the dynamics / temperature relaxation in a strongly coupled plasma is expected to be different from a non-strongly coupled plasma would be appreciated.

We itemized our models more clearly where the models are appropriate for weak, moderate and strong coupling. We have shown previously PoP 28, 062302 (2021) that coupled modes in strongly coupled plasmas give rise to oscillations in the time-evolving velocity distributions. Our point in this paper is that strong coupling gives rise to much faster decay rates, shown in Fig. 5.

4. I am missing a few references to recent experiments on the evolution of ultracold plasma in the strong-coupling regime such as [T. K. Langin et al., Science 363, 61 (2019)], [T. Kroker et al., Nat. Commun. 12, 596 (2021)], [M. A. Viray et al., Phys. Rev. A 102, 033303 (2020)] and [E. V. Crockett et al., Phys. Rev. A 98, 043431 (2018)] which the authors might want to consider.

These are excellent references and apologize that we did not include them in our earlier draft. They are all included now on line 68.

5. In the last paragraph of the introduction (p. 4) it is discussed that classical MD simulations are sufficient, however, earlier (p. 3) the authors point out that plasma models “must include effects of Fermi degeneracy [...]”. I am curious under which parameters Fermi degeneracy is expected to be of relevance, especially related to the experiment of the authors.

The Fermi degeneracy must be considered in HED plasmas. However, the electron temperature is much larger than $k_B T_F$ hence the system is not degenerate. We removed this confusing passage.

6. What do the authors mean by “same sign of charge” in the last paragraph of the introduction (p. 4)? Simply that the plasma consists of an ion-ion mixture? Why are the electrons (with negative charge) not relevant then?

We clarified this reference to same sign of charge in line 87.

7. The “Theoretical Considerations” (p. 4) assume a spatially homogeneous plasma while most experiments (including the conducted experiment) are limited to Gaussian shaped plasma. I would appreciate a short comment on the expected differences and why the authors consider a homogeneous plasma in the modelling.

We clarified this by adding three paragraphs starting in line 125 and also the new Fig. 1,

8. To remove ambiguities, $\bar{a}_{i,s}^3 = 3/47r_i n_{tot}$ should be formatted as $\bar{a}_{i,s}^3 = 3/(47r_i n_{tot})$ on p. 4.

We adjusted the equation.

9. On p. 4 the authors refer to Γ as the “strong coupling plasma parameter”. I’d rather call it coupling parameter because it distinguishes between strongly and weakly coupled plasma and is valid for both regimes.

We have removed the adjective “strong” in all cases.

10. The authors refer to a coupling parameter of $\Gamma = 3$. For clarity, it would be good to explicitly state which typical densities and ion temperatures this relates to in the experiment.

We have added a plot of Gamma, insert in Fig. 2. The densities and temperatures are noted in the caption to Fig. 4 and in the two paragraphs starting in lines 95 and 103.

11. On p. 5 the authors state that the "first model is found in the NRL Plasma Formulary". Please explicitly reference the model for clarity. I assume the authors mean Eq. (3-5) with the first model.

Starting in line 204, we clarified the models significantly. The first model, for example, was motivated by the NRL, but modified as discussed in the text. Hopefully this is more clearly presented in the text.

12. On p. 5 the authors refer to p. 33 in the NRL Plasma Formulary (2013). I find the relevant formulas one page later on p. 34 in the 2013 NRL Formulary. Additionally, I am confused about the units. The cross section in Eq. (5) does not seem to have the unit $[m^2]$. The collision frequency in Eq. (4) should have the unit of a rate $[1/s]$ but combining a cross-section in Eq. (5) $[m^2]$ and number density $[1/m^3]$ with the collision integral in form of the Debye length $[m]$ results in a unitless quantity. Even more confusing is the fact that the collision integrals for the other models in Eqs. (6,11,13) are dimensionless (unlike to $S = \lambda_{\alpha\beta}$ having the unit $[m]$).

We have updated the NRL Reference with the correct one. The reviewer is correct does not have the unit of cross section and we were wrong in calling so. Σ has units of $[m^3/s]$ and it is just a prefactor of physical constants and the collision integral Σ is dimensionless for all three models. The equation $\Sigma = \lambda_{\alpha\beta}$ refers to λ_{if} on page 34 in the NRL formulary and not to the Debye Length. We have removed it to avoid ambiguities.

13. Some physical sense of a negative Debye length (p. 5) or why it appears would be helpful.

As mentioned above $\lambda_{\alpha\beta}$ is the notation of the Coulomb logarithm in the NRL and not the Debye Length. The Coulomb logarithm becomes negative at strong coupling. We have modified the manuscript accordingly.

14. Eqs. (7-8) seem to differ from the treatment on hyperbolic trajectories in Ref. (25). Can the authors elaborate which model in Table I in Ref. (25) they refer to and how they arrive at the min. / max. distances in Eq. (7-8)? Moreover, are λ_1 and λ_2 equal to λ_{α} and λ_{β} ?

We have rewritten the discussion of the Models. Each model has its own sub-section. Because none of the models are directly equivalent to the published works, we have clarified the modifications we made to account for the mass ratio and the collision terms.

15. On p. 6 the authors state that by using Ref. (25) the NRL model is generalized to ion-ion collisions. Does this mean they are not included at all in the NRL approach?

This portion of the manuscript has been heavily rewritten to describe the models much more clearly and in much more detail. We hope that this question, and many others, are now answered in the new text.

16. Is x_{α} in Eq. (9) the concentration n_{α}/n_{tot} ? And is Eq. (9) derived from Eq. (35) in Ref (18)? If so, can the authors briefly elaborate how they arrive at the result or if further assumptions / approximations are made? Again, in Eq. (9) the species are not referred to as α and β but 1 and 2 with α as dummy index in the sum. That must be consistent in the manuscript.

This result was developed in a previous paper. Briefly, the logic that we bring from that work is that Debye-Hückel theory predicts that the screening cloud collapses at low temperature. However, we know that this can't persist below the interparticle spacing. From observations of the properties of $g(r)$ a lower limit was set for the screening length. Here, we adopt that idea and use it in the (hyperbolic) Coulomb logarithm. We have added some comments that will help future readers so that one does not need to refer to the original paper.

17. What do n and m in $n = m = 1$ refer to on the bottom of p. 6? We have provided an equation for $K_{11}(g)$ and removed the line with $n = m = 1$.

18. In Fig. 1(a) the experimental setup is presented. While there are filters on the PMT branch, the light to the ICCD camera does not seem to be filtered. However, Fig. 1d presents Ca⁺ fluorescence selectively. Is this achieved by solely shining in the Ca⁺ fluorescence laser? The 250 μ m pinhole sets a boundary for the extent of the probed central part, however, I am missing the width of the cylindrically focused laser light along the imaging axis to get an idea of the area the signal is integrated over. Is the typical PMT signal in Fig. 1(b-c) for Ca⁺ or Yb⁺ ions? I am missing a discussion of Fig. 1(d). The subfigure needs a scale bar to get an idea of the size of the plasma. Moreover, I am curious about the spatial structure visible in the image. Why is there a "wing" left to the Gaussian spot in the center? As the authors seem to be able to take snapshots of the spatial evolution of the central dual species plasma region in parallel to the PMT signals, I would be happy to see a short series of images on how the plasma evolves in real space and if this expansion is consistent with the measured velocity profiles.

We added the filter in the figure. The thickness of the probe laser beam sheet (the rms size) is 0.146mm (or the Gaussian waist is 0.292 mm). We added a scale bar to Fig 1d. The evolution of the plasma size and structure are described in Fig. 1 and in the text for several paragraphs starting on line 109. In the figure caption to Fig 8 we noted that (c) is Ca⁺ fluorescence.

19. Is the peak density in the MOT n_0 on p. 7 referring to the total atomic density or the peak density for each species?

Peak density of each species. Hopefully this is clarified in the several place where this appears.

20. I suggest including a reduced overview level scheme of Ca and Yb atoms and all involved wavelengths (MOT and photoionization), which would make following the experimental procedure a lot easier. The same holds for the Ca⁺ and Yb⁺ ions involving all probe and optical pumping lasers.

We added a new Fig. 6.

21. I am missing the wavelengths used for photoionization in the manuscript. Moreover, does an excited state fraction in the MOT alter the photoionization scheme (e.g. different electron temperatures resulting from a non-resonant two-photon ionization process)? If so, do the authors know the excited state fraction in the MOT?

We added Fig. 6 which clarifies these wavelength questions. The MOT atoms are ionized in a two-photon, two-color resonant process. We turn off the MOT lasers prior to ionization so the excited state fraction at the time of ionization is zero.

22. The authors state that the electron temperature is around 96 K. Does this mean the electron temperature is experimentally tunable? If so, in what range? And does a modified electron temperature change the initial ion temperature?

The electron temperature is determined by the energy of the ionizing pulse above the continuum (See Fig. 6). We mention the electron temperature range in the paragraph starting on line 103. The lower limit in principle is limited by the laser linewidth to about 1 K. However, three-body recombination rapidly increases the electron temperature when the initial T_e is very low. In practice, when $\kappa = \bar{\alpha} n_s / \Lambda T_e$ is less than about 0.5, recombination can be ignored. The ion velocity due to photon recoil is 2 cm/s (negligible). At an electron temperature of 100 K, ion velocity due to electron recoil is 0.5 m/s for Ca and 0.1 m/s for Yb, corresponding to mK temperatures.

23. Are the initial plasma densities independently tunable for each species in the experiment?

Yes, by independently turning off the MOT laser beams we can let each neutral atom cloud expand independently. See new figure 7 in paper.

24. What is the pulse duration of the ionization laser pulse? What is the duration of the probe pulse?

Ionization pulses are 5-9 ns (line 103). The probe lasers are on for the duration of the plasma expansion. As discussed in the paragraphs starting in lines 339 and 345, using pulsed probe lasers does not change the results.

25. Do the authors know how sensitive the experiment / expansion dynamics is to electric stray fields and to what degree are external fields under control?

The vacuum chamber is grounded, all non-conducting surfaces are several inches from the plasma, and no electric fields are applied. Weak stray fields, if they are present, would only influence the very late-time evolution of the plasma and not the early time evolution considered in this work.

26. On p. 7 it is stated that the authors "verify that optical pumping is a negligible source of error". A brief sentence how this is achieved would be appreciated.

This discussion is added in the paragraphs starting in lines 339 and 345.

27. On p. 8 the authors discuss that disorder induced heating (DIH) increases the temperature to near 1 K. How does a flat atom pair distribution function relate to a significant Coulomb coupling of $\Gamma = 3$ (p. 4) where the ions are expected to form spatial correlations?

Yes, the atomic distribution function $g_a(r) = 1$ is initially flat. Immediately after the plasma is formed it inherits this $g_p(r, 0) = g_a(r)$ – at that moment, the Coulomb coupling is extremely high (many tens of thousands) in the sense that the temperature is very low. However the ions are in the wrong locations. The $g_a(r)$ is inconsistent with a strongly-coupled plasma. In the DIH process the plasma ions move to form a new $g_p(r, t)$. Because heating occurs during DIH, we end up with a plasma with $\Gamma < 3$ at an equilibrium $g_p(r, \infty)$ consistent with that lower coupling. Two new sentences and a new reference have been added to clarify this process. See line 146.

28. What is a typical runtime (minutes / hours / weeks) of the MD simulation and what computer / cluster has been used to carry out the simulation?

We presented this in the paragraph starting on line 386.

29. On p. 8 it is stated that the Yb ions are created first, and Ca is ionized 40 ns later. Why is the ionization not performed simultaneously? In general, a sketch of the experimental sequence would be very helpful. How long is the MOT phase, ionization sequence etc.?

Experimentally, the 390 nm laser pulse used to ionize Ca will also ionize any Yb atoms in the excited state. The 40 ns delay is used to ensure that no Yb atoms are ionized by the Ca ionization pulses. Any "cross-ionization" by the wrong laser pulses would result in a very high and uncontrolled electron temperature. This is explained in line 324 in the manuscript.

30. Please explicitly state the experimental electron temperatures and densities used for deriving the value of α on p. 9 for clarity.

We added this information in line 393.

31. Am I correct that the MD simulation calculates the dynamics of the $N = 50,000$ ions (p. 10) but not explicitly the trajectories of the involved electrons?

Yes. We simulate the time evolution of $N = 50,000$ ions without electrons. The presence of the electron background is incorporated in the screening parameter α and λ_{TF} . We have modified the manuscript to elucidate this point. This is clarified in line 366.

32. On p. 11 it is stated that the MD velocity distribution is convolved with a Lorentzian distribution. What is the width of the Lorentzian distribution, and does it relate to the experimental linewidth of the laser? Connected to this: Please state the experimental velocity resolution resulting from the laser linewidth. The unconvolved simulation result would show the more fundamental dynamics of the temperature time-evolution. The authors might consider showing the raw results of the simulation in an inset or figure in the Supplementary Material.

The laser linewidth is less than 1 MHz. The velocity resolution is set by the atomic transition's Lorentzian HWHM of 11 MHz, equivalent to 4 m/s. The typical Gaussian rms width is 20 m/s. The convolution has a minor effect on the rms velocity extracted from the MD. Because this question was treated in a recent publication by Killian's group, we have discussed the effect and referenced Killian's paper. This information is now in lines 404 – 411 in the manuscript.

33. What is the origin of the oscillations in temperature within the first 00 ns in Fig. 3 and is the oscillation frequency connected to the plasma frequency?

These are kinetic energy oscillations initiated in the DIH process. The frequency is given by the ion plasma frequency. They are discussed in references such as Phys. Rev. Lett. 96, 16 001 (2006) and Phys. Rev. E93, 023201 (2016). We added this to the caption of Fig. 2.

34. On p. 11 the authors mention the influence of the density gradient in the probed region. What is the width of the light sheet (i.e. the probed region) and the density modulation over this region for both experimental regimes? Why is the density and size of the Yb atoms modified stronger (one order of magnitude for the density) compared to the Ca atoms (< 20%) in Fig. 3?

The rms widths of our plasmas are > 500 μm . The thickness of the probe laser beam sheet (the rms size) is 0.1 mm, as mentioned now on line 337. The limits in space and time for maintaining density variations of less than $\pm 10\%$ are listed on line 132. The density ratios were chosen so that in one case Yb was a minor component and in the other Ca was a minor component.

35. On p. 12 the authors mention non-binary collisions as reason for the overprediction. I am wondering what the influence of three-body recombination (TBR) into Rydberg states is. Can the authors estimate the TBR rate for the dual species plasma?

Three body recombination rate goes as $T_e^{-9/2}$. As mentioned previously in this reply, we keep \bar{n} less than 0. to avoid recombination into Rydberg states. We added a comment and reference on line 412.

36. I am missing a few details in Fig. 4. The comparison to the theoretical models in Fig. 4 seems to start after 1 μs or do all models fall onto the MD curve for shorter times? Why are the models not compared to the MD data for shorter evolution times? Which equation is used for the experimental fit from which the parameter $\nu = 2.52$ MHz is extracted? Can we deduce a cross section or collision integral from that?

The Hermite analysis presented in the Simulations section was used to find the time at which the two species had reached a Maxwellian velocity distribution (after about 1 μs). The comparison with theory was started at this time. This is because all three models assume Maxwellian velocity distributions. We have modified the manuscript to make this point clearer. We removed the fit from the data. In principle, a cross section could be extracted from the fitted decay rate, depending on the reliability of the underlying theory. This is a classic inverse problem that probably is not well posed in the present work. You have a rate and want to deduce a cross section that is inside an integral.

37. The authors claim a strongly coupled binary ionic mixture. The coupling parameter is given as $\Gamma = 3$, however, depends on the time-evolution of the density and temperature. Can the authors estimate how Γ evolves with time in the experimental regimes?

We have added a plot in the new Fig. 2 showing the time-evolution of Gamma.

Reviewer 3

We express appreciation to the reviewer for their comments. We excerpt direct requests in black and indicate our response in blue.

1. The authors claim that elaborating through the UNP plasma specificities allow them to perform a quantitative experience-simulation meaningful match while securing transferability of their conclusions to more mundane plasmas:ICF-or WDM- like....This point should be strengthened more convincingly.

We added a new paragraph starting on line 67 addressing this point. We have included a new reference to Stanek et al. on ionic pair interactions, beginning on line 288.

2. The three models confronted on pp.5-6 should be given more details about their physics content. It is not sufficient to drive the reader to the third author former theory works.

We have rewritten that portion of the manuscript extensively. We walk the reader through each model in more detail, with more insights into what physics is added at each step.

3. Editing remarks:

- (a) p.5 line 8/bottom....Lamda with suffixes alpha beta (CL) should be explained We have removed $\Sigma = \lambda\alpha\beta$ to avoid ambiguities

- (b) p.6 line 7/top....Lamda TF needs also to be detailed in view of its significance We added a reference and words in lines 166 and also 365–369

- (c) p.14 Ref.14....-Taylor,Richtmyer-Meshkov....

- (d) p.20 Ref.53....BGK model...

- (e) p.20 Ref.55....Yukawa

We have corrected the References.

- (f) p.26 Caption Fig.3 apparently Sigma Ca and Sigma Yb are not introduced neither in text nor here Those symbols are defined starting in line 98.

REVIEWER COMMENTS

Reviewer #1 (Remarks to the Author):

The authors have addressed all of my points raised in my previous report, with appropriate modifications made to the manuscript. The revised manuscript can now be accepted for publication in Nature Communications.

Reviewer #2 (Remarks to the Author):

The authors have adequately addressed my comments in this revision and I am happy to recommend publication of the manuscript in Nature Communications. Moreover, I congratulate the authors to their results!

Minor comments:

- p. 11: λ is not introduced.
- p. 14: The influence of coupled modes is discussed by giving specific equations, however, no quantitative comparison to the data is made using these equations. The authors might consider discussing the influence only brief and qualitatively.

Philipp Wessels-Staarmann

Reviewer #3 (Remarks to the Author):

The authors have dutifully reacted to every referees remarks

The main interest of this presentation concerns the metrology dedicated to non-equilibrium strongly coupled plasmas. The very physics output about temperature-relaxation is not that impressive.

However, the huge amount of excellent work displayed in simulations as well as in the experimental procedure

together with the significance of the topic for the strongly coupled plasma and ICF communities might justify publication in Nat.Comm.

The authors claim the non-validity of the considered models. So, they could have proposed their own models stemming from their simulation-experiment confrontation.

They could refer to the famous British statement: All models are wrong, some are useful.

We express our appreciation to the editor and referees for their comments on our manuscript. To address editorial requests, a few changes to the manuscript were required. Some clarification was requested regarding Ref. [13] and its relationship to the present work. That clarification necessitated changes in Figures 3 and 4, expansion to the two-temperature analysis, moving some material to Supplementary Information, and the addition of Ref. [14]. These changes are shown in blue type.

Comments from the Reviewers

Reviewer #1: Thank you for the congratulatory remarks.

Reviewer #2: Thank you for your congratulatory remarks.

1. We moved the definition of $\lambda_{1,2}$ from the Methods section to the text following Eq. (15).
2. The influence of mass ratio on coupled modes is mentioned briefly in line 314.

Reviewer #3:

Thank you for your compliments on our work. We agree that an accurate coupled mode theory would be a significant and herculean accomplishment. It is an area of ongoing research.